# Spatial Interpolation of Soil Temperature and Water Content in the Land-Water Interface Using Artificial Intelligence

**Hanifeh Imanian** [1,*], **Hamidreza Shirkhani** [1,2], **Abdolmajid Mohammadian** [1], **Juan Hiedra Cobo** [2] **and Pierre Payeur** [3]

1  Department of Civil Engineering, University of Ottawa, Ottawa, ON K1N 6N5, Canada
2  National Research Council Canada, Ottawa, ON K1A 0R6, Canada
3  School of Electrical Engineering and Computer Science, University of Ottawa, Ottawa, ON K1N 6N5, Canada
*  Correspondence: himania3@uottawa.ca

**Abstract:** The distributed measured data in large regions and remote locations, along with a need to estimate climatic data for point sites where no data have been recorded, has encouraged the implementation of spatial interpolation techniques. Recently, the increasing use of artificial intelligence has become a promising alternative to conventional deterministic algorithms for spatial interpolation. The present study aims to evaluate some machine learning-based algorithms against conventional strategies for interpolating soil temperature data from a region in southeast Canada with an area of 1000 km by 550 km. The radial basis function neural networks (RBFN) and the deep learning approach were used to estimate soil temperature along a railroad after the spline deterministic spatial interpolation method failed to interpolate gridded soil temperature data on the desired locations. The spline method showed weaknesses in interpolating soil temperature data in areas with sudden changes. This limitation did not improve even by increasing the spline nonlinearity. Although both radial basis function neural networks and the deep learning approach had successful performances in interpolating soil temperature data even in sharp transition areas, deep learning outperformed the former method with a normalized RMSE of 9.0% against 16.2% and an R-squared of 89.2% against 53.8%. This finding was confirmed in the same investigation on soil water content.

**Keywords:** artificial intelligence; deep learning; RBF networks; rail infrastructure; sharp transition; soil temperature; spatial interpolation; water content

## 1. Introduction

Spatial interpolation techniques are widely used in many areas of science and engineering such as the mining industry, environmental sciences, relatively new applications in the field of soil science and meteorology, and most recently, public health.

In all areas, the general context is that some phenomenon of interest is occurring in the landscape, for example, metal content in a mine, oil pollution in coastal waters, soil moisture level, groundwater depth, rainfall, electromagnetic level, etc. Since exhaustive studies are not feasible, the phenomenon is usually characterized by taking samples at different locations. Interpolation techniques are then used to produce estimations for the unsampled locations [1].

There are two main categories of interpolation techniques: deterministic and stochastic or geostatistic. Deterministic interpolation techniques create surfaces from measured points, based on either the extent of similarity, such as inverse distance weighted (IDW), or the degree of smoothing, as with radial basis functions (RBF). Natural neighbors, trend, and spline fall under the deterministic category. On the other hand, geostatistical interpolation techniques such as kriging utilize the statistical properties of the measured points. Stochastic methods quantify the spatial autocorrelation among measured points and account for the spatial configuration of the sample points around the prediction location [2].

There are two kinds of deterministic interpolation methods. An interpolation approach that estimates a value equal to the measured data at a sampled location is known as an exact interpolator. In the inexact method, the interpolator does not force the resulting surface to pass through the measured values, so the estimated values can be different from the measured data. This approach can be used to avoid sharp peaks or troughs in the output surface. IDW and RBF are exact interpolators, while global and local polynomial and kernel interpolation are inexact.

Numerous studies have focused on the applicability and accuracy of deterministic and geostatistical interpolation methods to estimate climatic variables.

Wu et al. (2016) compared different approaches for spatial interpolation of soil temperature including IDW, ordinary kriging (OK), multiple linear regression, regularized spline with tension, and thin plate spline. They concluded that the thin plane spline gave the best performance [3]. Adhikary and Dash (2017) compared the performance of two deterministic, IDW and RBF, and two stochastic, OK and universal kriging (UK), interpolation methods to predict the spatiotemporal variation of groundwater depth. The analyses revealed that the UK methods outperformed the others [2]. Mohammadi et al. (2017) used some interpolation methods such as IDW, kriging with constant lapse rate and gradient inverse distance squared with lapse rate determined by classification and regression tree to interpolate the bias-corrected surface temperature forecasts at neighboring observation stations to any given location [4]. Wang et al. (2017) compared different spatial interpolations of IDW, kriging, spline, multiple linear regression and geographically weighted regression models for predicting monthly near-surface air temperature. The assessment result indicated that the geographically weighted regression model is better than other models [5]. Rufo et al. (2018) used IDW, spline, and OK to represent the electric field levels in a village [6]. Amini et al. (2019) mapped monthly precipitation and temperature extremes in a large semi-arid watershed using six spatial interpolation methods: IDW, Natural Neighbor, Regularized Spline, Tension Spline, OK and UK [7]. Kisi et al. (2019) assessed kriging performance in interpolating long-term monthly rainfall [8]. Ahmadi and Ahmadi (2019) studied the spatiotemporal distribution of sunshine duration using the empirical Bayesian kriging (EBK) method [9]. Zhu et al. (2019) used the kriging interpolation method in order to study the spatiotemporal changes in farmland soil moisture at various soil depth levels [10]. Yang and Xing (2021) applied six interpolation methods of IDW, RBF, diffusion interpolation with a barrier (DIB), kernel interpolation with a barrier (KIB), OK and EBK to estimate different rainfall patterns and showed that the KIB method had the highest accuracy [11].

However, conventional interpolation methods have some limitations. They make many assumptions, can be computationally demanding, and it may not be easy to define a geostatistical model for data that cannot easily be transformed into normality [12].

Currently, with the increasing use of machine learning (ML), it can be considered a promising alternative to traditional algorithms for spatiotemporal interpolation. ML relies on the relationship between dependent variables and independent parameters; thus, stronger correlation leads ML models to produce remarkably more accurate results. In recent years, remote sensing helps ML in spatiotemporal interpolation significantly since it gives a boost in collecting the required data as the ML models' input database.

Many researchers have used ML methods in spatiotemporal interpolation in different fields. Appelhans et al. (2015) evaluated several ML approaches, including support vector machine (SVM), neural network, k-nearest neighbors (kNN), random forest (RF), gradient boosting, etc. for the interpolation of monthly air temperature. They considered a combination of kriging and ML models as the best solution [13]. Hengl et al. (2018) introduced an RF for spatial prediction, which uses buffer distance maps from observation points as covariates. They showed that adding geographical proximity effects into the prediction process improved prediction [14]. da Silva Junior et al. (2019) evaluated ML strategies represented by RF and an RF variation for spatial estimation and conventional strategies such as IDW and OK to spatially interpolate evapotranspiration data [15].

Guevara and Vargas (2019) trained the kernel weighted nearest neighbors method to downscale annual soil moisture [16]. Mohsenzadeh Karimi et al. (2020) compared different ML methods such as RF and SVM and geostatistical models to predict monthly air temperature. The comparisons demonstrated the superiority of SVM and RF models over other models [17]. Xu et al. (2020) proposed a machine learning-based geostatistical downscaling method for downscaling land surface temperature, which integrates the advantages of RF and area-to-point kriging methods [18]. Cho et al. (2020) improved the spatial interpolation accuracy of daily maximum air temperature by using a stacking ensemble model consisting of linear regression, support vector regression (SVR), and RF [19]. Sekulic et al. (2020) introduced a novel spatial interpolation method that considered greater values to the nearest observations and distances to the nearest observations. They compared their method with other widely-used methods such as IDW, kriging, and RF and showed its superiority [12]. Chen et al. (2021) proposed a downscaling method based on RF which considered the spatial autocorrelation of precipitation measurements between neighboring locations and was able to estimate precipitation with high resolution [20]. Leirvik and Yuan (2021) applied RF for spatial interpolation of solar radiation observations and compared their results with OK and regression kriging (RK) and demonstrated a relatively better performance of RF [21].

A recent review of the literature shows that although deterministic interpolation techniques are still popular methods, the advantages of ML methods are so impressive that they can be a substitutional way to the convolutional methods, mainly in cases that failed to produce reasonable results.

When the temperature reaches a certain threshold, the rails are susceptible to stretching and deforming. To ensure safety, trains must run at a reduced speed. Slow order is a normal procedure when the railway reaches a certain temperature. It may delay train arrival and departure. The main motivation for conducting this study was to map soil temperature in proximity to a railroad as a key component of the health monitoring system for rail infrastructure. The further step can be applying the models to predict rail track temperatures and evaluating their performance.

This paper presents the evaluation results of machine learning-based algorithms against conventional strategies for interpolating soil temperature data from a region with an area of 1000 km by 550 km in southeast Canada that was recorded in May 2021. In this regard, two methods of radial basis function neural networks and the deep learning approach were used to estimate soil temperature along a railway after some deterministic spatial interpolation methods failed to interpolate gridded soil temperature data on the desired railroad.

The rest of the paper is organized as follows: Section 2 describes the used approaches and their hyperparameters, the considered case study and used error metrics. Section 3 presents the evaluation results, which are summarized and discussed in Section 4. Lastly, Section 5 includes some concluding remarks and suggestions for future study.

## 2. Materials and Methods

### 2.1. Study Area and Dataset

The main study area of the project was the Quebec City—Windsor Corridor (see Figure 1). This passenger train service has the heaviest passenger train frequency in Canada. The region extends between Quebec City (46.8° N, 71.2° W) in the northeast and Windsor (42.3° N, 83.0° W), Ontario, in the southwest, spanning 1150 km. With more than 18 million people, it contains approximately half of the country's population and three of Canada's four largest metropolitan areas. It is the most densely populated and heavily industrialized region of Canada.

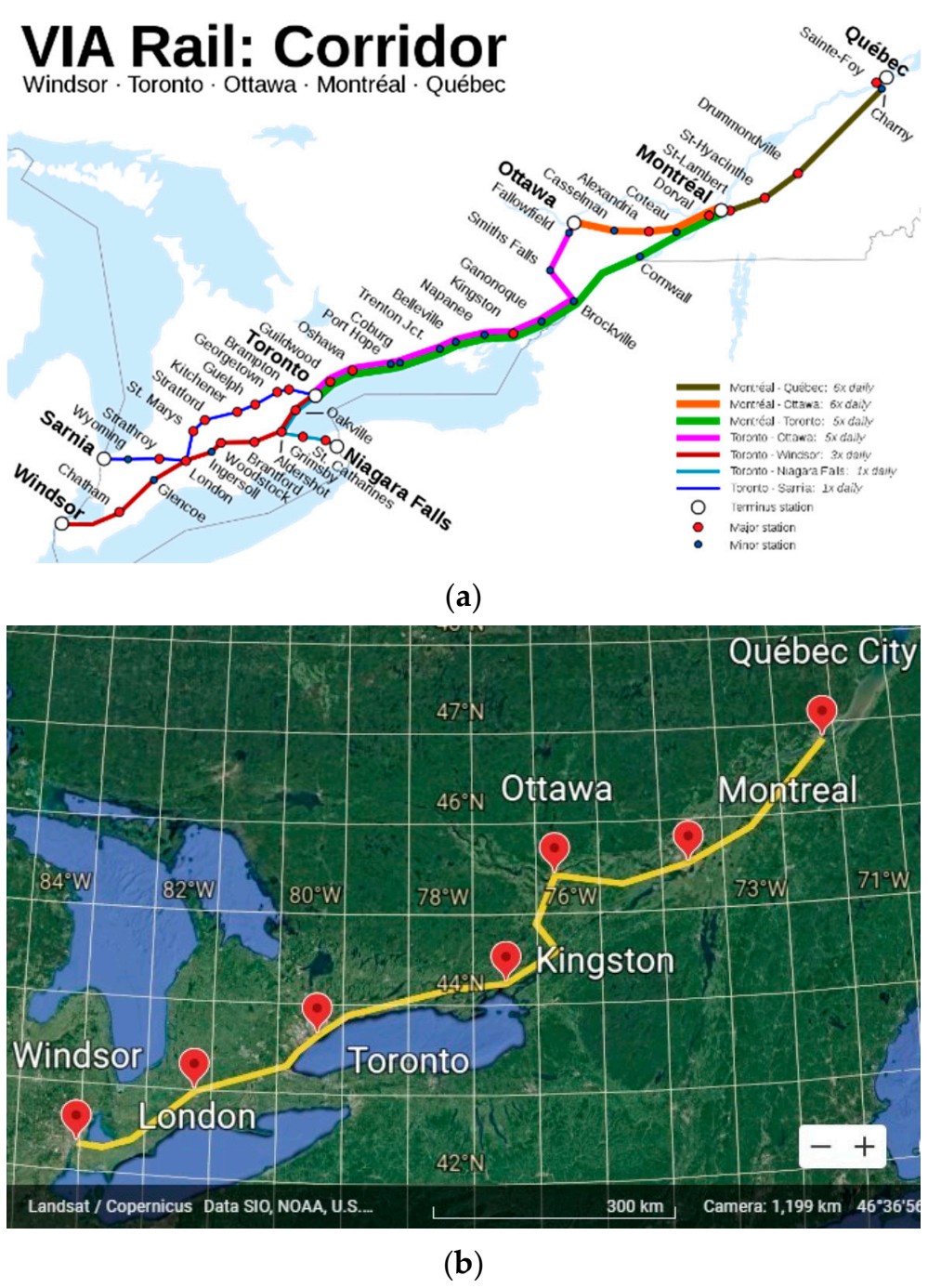

**Figure 1.** Study area (**a**) Quebec City-Windsor Corridor map [22] (**b**) main rail stations [23].

The climate data used herein were obtained from the database of the Meteorological Service of Canada, which is a division of Environment and Climate Change Canada and primarily provides public meteorological information and weather forecasts. The database contains data from analysis systems along with output from many of the Canadian Meteorological Centre's Numerical Weather Prediction models. It provides estimates of a large number of atmospheric, land and oceanic climate variables in a gridded-base format. The dataset fields are made available on a 935 by 824 polar-stereographic grid covering North America with a 10 km resolution at 60° N. The soil temperature data (in Kelvin) and soil moisture data (in kg/m$^2$) were downloaded from the freely accessible website of Environment and Climate Change Canada, which is the department of the Government of Canada responsible for coordinating environmental policies and programs

(https://weather.gc.ca/grib/grib2_reg_10km_e.html, accessed on 15 May 2021). Both soil temperature and soil moisture variables were collected 0–10 cm below ground level.

The soil temperature values in the considered study area are demonstrated graphically in Figure 2. The maximum value of the soil temperature dataset is 296 K, and the minimum value is 270 K. The mean, median and mode are 289 K, 291 K and 292 K, respectively, so the data skewed to the left. This information has a 6 K standard deviation.

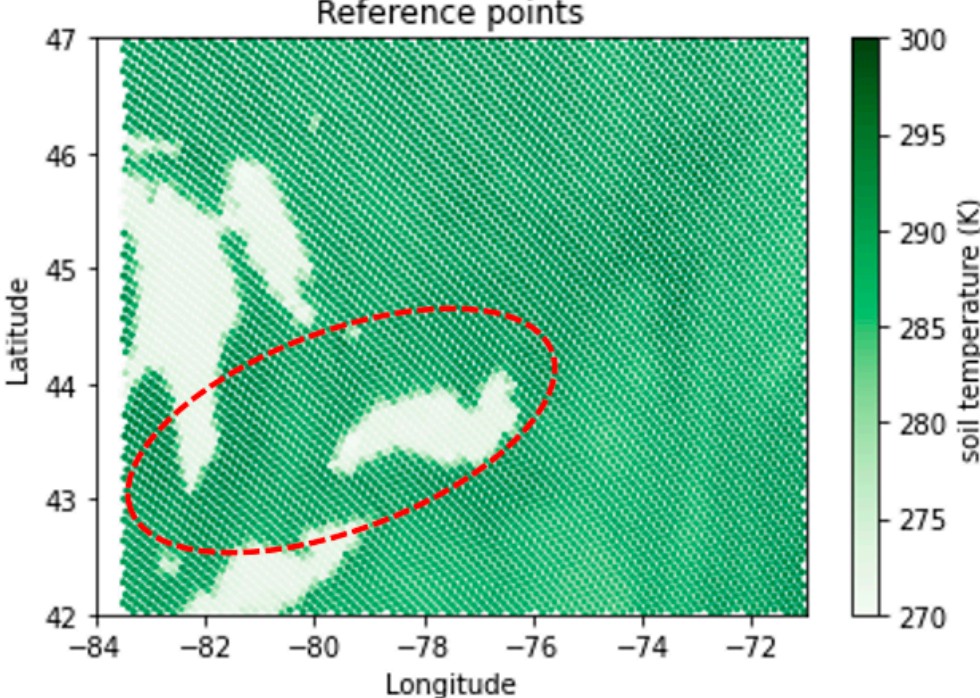

**Figure 2.** Graphical demonstration of gridded values of soil temperature in the study area, reported by Environment and Climate Change Canada (dashed circle shows the area of interest).

### 2.2. Description of Applied Methods

In this section, the deterministic interpolation methods used in the present study are summarized. Two used AI approaches are then reviewed, including RBF networks and deep learning neural networks.

In an earlier study by Imanian et al. (2022), a wide range of AI approaches from classic regressions of Ridge, Lasso and Elastic net; to well-established methods of Nearest Neighbors, RF, gradient boosting and SVM to more advanced AI techniques, such as MLP and deep learning, are taken into account and their performance in soil temperature prediction was compared in a comprehensive study [24]. The outcome of this investigation was that deep learning showed the best performance in predicting soil temperature with the highest correlation coefficient and lowest error metrics. Therefore, in the present study, the interpolation models were compared with the best model of the above-mentioned study, the deep learning technique.

#### 2.2.1. Deterministic Interpolation

The idea behind deterministic interpolation methods is to assign values to ungauged locations according to the surrounding measured points based on specified mathematical functions that determine the smoothness of the created surface. In the present study, different linear and nonlinear types of spline method were used.

The spline set of interpolation polynomials is part of a general class of interpolation methods that are referred to as piecewise polynomial interpolations. The 3D spline method bends a surface that passes through the known values while minimizing the total curvature of the surface. Thus, this method usually applies for interpolation in cases dealing with gently varying surfaces.

The advantages of this method are the simplicity of calculation, numerical stability, and smoothness of the interpolated surface. On the other hand, this method has some limitations in representing the sudden changes in gradient or extreme differences in values for close points. This is because the spline method uses slope calculations (change over distance) to figure out the shape of the surface [25].

As mentioned above, the spline method fits a mathematical function to a specified number of nearest input points. In the present study, three methods of linear, cubic spline, which is a third-order polynomial, and quintic spline, which is a polynomial of degree five, are used for interpolation. The function would be in the form of $(X, Y, Z)$, with $(X, Y)$ being the geographical location of latitude and longitude and $Z$ being the soil temperature. For the linear method, the minimum points required along the interpolation axis is 4. The number of points used in the calculation of each interpolated cell in the cubic and quintic spline methods was 16 and 36, respectively.

### 2.2.2. Radial Basis Function Neural Networks

A radial basis function network is the basis for a series of networks known as statistical neural networks that use regression-based methods in contrast to conventional neural networks [26].

The radial basis function network is a feedforward artificial neural network that uses radial basis functions (RBF) as activation functions. RBF networks (RBFN) have many applications, such as function approximation, interpolation, classification, and time series prediction. An RBFN model comprises three layers: an input layer, a hidden layer with a nonlinear RBF activation function, and a linear output layer [27]. Since the original architecture of the RBF network always has only one hidden layer, the convergence of optimization is much faster than other neural networks and requires less time to reach the end of training.

Apart from the number of hidden layers and linearity in the output layer, there are some other differences between RBF networks and multilayer networks, including (i) the RBFN activation function is based on the distance between the input vector and center, while the activation function of a multilayer network is based on the computation of an inner product, and (ii) the nodes of a multilayer network all have the same activation function; which is not the case for RBFN [28].

Commonly, RBF networks are calibrated using a three-step procedure. The initial step concerns the choice of center vectors for the RBF neurons in the hidden layer. The following step includes a determination linear model whose weights are associated with the outputs of the layers. The third step concerns the adjustment of all the RBF network parameters by using backpropagation iteration. Backpropagation, or backward propagation of errors, is a process that involves taking the error rate of a forward spread and feeding this loss backward through the neural network layers to fine-tune the weights. Backpropagation is the essence of neural net training.

To illustrate the working flow of the RBFN, suppose we have a dataset which has N patterns of $(x_p, y_p)$, where $x_p$ is the input of the data set and $y_p$ is the actual output.

The output of the $i$ th activation function $\phi_i$ in the hidden layer of the network can be calculated using the following equation based on the distance between the input pattern $x$ and the center of RBF unit $i$ [29].

$$\phi_i(\|x - c_i\|) = exp\left(-\frac{\|x - c_i\|^2}{2\sigma_i^2}\right) \quad (1)$$

Here, $\| \ \|$ is the Euclidean norm, $x$ is the input vector, $c_i$ and $\sigma_i$ are the center and width of the hidden neuron $i$, respectively.

The computations in the output layer are performed just like a standard artificial neural network, which is a linear combination between the input vector and the weight

vector. The output of the node *k* of the output layer of the network can then be calculated using the following equation.

$$y_k = \sum_{i=1}^{n} \omega_{ik}\phi_i(x) \tag{2}$$

where $\omega$ is the weight connection, $\phi_i$ is the *i* th neuron's output from the hidden layer, *y* is the prediction result, and *n* is the number of neurons in the hidden layer.

In the current model, there were 50 neurons in the only hidden layer and the maximum iteration was set to 700. The tuning results of hyperparameters are presented in Table 1. The selected values led to a higher R-squared.

**Table 1.** Tuning of hyperparameters.

| Maximum iteration (RBFN) | 100 | | 500 | 700 | | 1000 |
|---|---|---|---|---|---|---|
| R-squared | 0.54655 | | 0.54378 | 0.54957 | | 0.54641 |
| Neurons in hidden layer (Deep learning) | 300 | 500 | 300, 30 | 300, 100 | 500, 30 | 500, 100 |
| R-squared | 0.83668 | 0.84645 | 0.85651 | 0.87846 | 0.88696 | 0.89011 |

### 2.2.3. Deep Learning

A deep neural network is a collection of neurons organized in a sequence of multiple layers where neurons receive the neuron activations from the previous layer as input and perform a simple computation (for example, a weighted sum of the input followed by a nonlinear activation). The neurons of the network jointly implement a complex nonlinear mapping from the input to the output. This mapping is learned from the data by adapting the weights of each neuron using a technique called error backpropagation [30].

The most crucial step in the learning process of a neural network is backpropagation, which propagates errors in the reverse direction from the output layer to the input layer and updates the weights in each layer. Since the number of hidden layers in a neural network model is directly related to its ability, backpropagation in a large network with many hidden layers can be computationally expensive and fail to handle the nonlinearity in the data [31].

Deep learning is not only a machine learning technique that uses deep hidden layers in a neural network but also has technically improved the backpropagation algorithm [24].

The deep learning method is assisted by some numerical methods that are beneficial for the training of the deep neural network and preventing the vanishing of gradients in the hidden layers. It also solves the overfitting problem by training only some of the randomly selected nodes rather than the entire network. By using some other algorithms and employing GPU, the deep learning technique can reduce high training times, which are due to heavy calculations.

Different architectures have been considered to tune the number of hidden layers and neurons and the calculated R-squared is presented in Table 1. It was found that considering two hidden layers with 500 and 100 neurons in the first and second hidden layers had the best performance.

The number of neurons in each hidden layer that controls the representational capacity of the network was set to 500 and 100, accordingly.

An epoch is the number of times that the neural network reads the entire training dataset during the training process. In the current study, two values of 500 and 800 were considered for the epoch. Trial and error showed that increasing epochs did not lead to better results, so the epoch was set to 800.

The optimizer used to train the network was the widely-used algorithm of Adam, which is a stochastic gradient-based optimizer [32]. The activation function for all hidden layers was the rectified linear unit (Relu), which passes the maximum of the variable and

zero and is recommended widely. This function controls the non-linearity of individual neurons [33]. The considered loss function was the mean square error (MSE).

## 2.3. Methodological Overview

As detailed earlier in Section 2.1, an area of approximately 1000 km by 500 km in southeast Canada called "the corridor" was studied. The input dataset for interpolation included 6640 gridded data of soil temperature and related coordinates. We used three types of spline methods including linear, cubic, and quintic, and the two methods of the RBF neural network and deep learning as machine learning methods to spatially interpolate the input data along the railroad. The railroad was mapped with 1300 pairs of longitude-latitude coordinates with an interval of 1 km. The study area with reference points and interpolation points is illustrated in Figure 3.

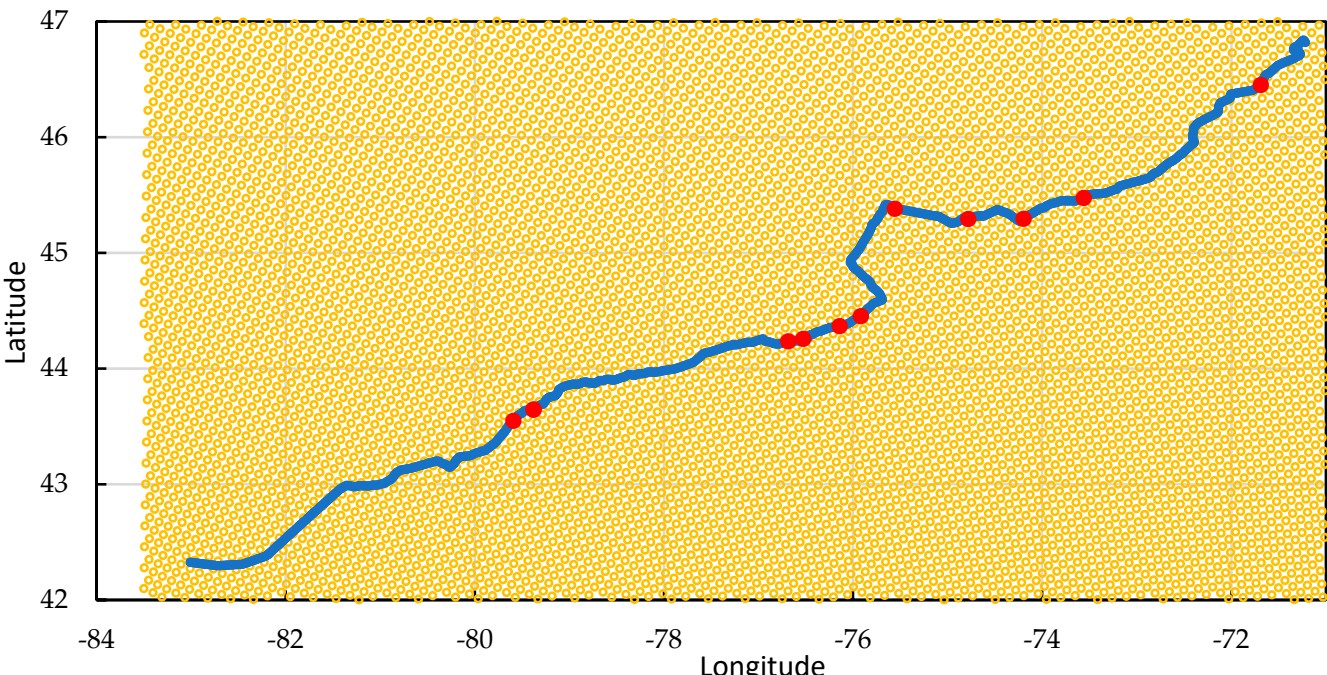

**Figure 3.** Location of 6640 reference points (orange dots), 1300 interpolation points (blue dots) on the railroad, and 12 evaluation points (red dots).

The input data (6640 reference points) were randomly split into two parts. This means data were shuffled first, then 65% of the data was devoted to the training set, and the remaining 35% of the data was set aside for validation. The railroad points (1300 interpolation points) were used for the testing phase.

In the next step, the considered interpolation technique was fitted to the training subset, and the estimation was then made on the validation dataset. The accuracy of interpolation on the validation set was calculated based on several error metrics, as described in Equations (3)–(14). If the error analysis was acceptable, according to the optimal value, the fitted interpolation model was applied to the test set to calculate the soil temperature interpolated along the railroad.

The outputs of the developed interpolation models were compared to facilitate the evaluation of each model's performance. Some error metrics, including maximum residual error (MaxE), mean absolute error (MAE), mean square error (MSE), root mean square error (RMSE), normalized root mean square error (NRMSE), scatter index (SI), mean absolute percentage error (MAPE), bias, coefficient of determination (R-squared), Nash-Sutcliffe

efficiency coefficient (NSE), variance accounted for (VAF), and Akaike information criterion (AIC) are used to assess the modeling accuracy, which is defined as:

$$MaxE = Max(y_{obs} - y_{calc}) \quad \text{Optimal value} = 0 \tag{3}$$

$$MAE = \frac{\sum |y_{obs} - y_{calc}|}{n} \quad \text{Optimal value} = 0 \tag{4}$$

$$MSE = \frac{\sum (y_{obs} - y_{calc})^2}{n} \quad \text{Optimal value} = 0 \tag{5}$$

$$RMSE = \sqrt{\frac{\sum (y_{obs} - y_{calc})^2}{n}} \quad \text{Optimal value} = 0 \tag{6}$$

$$NRMSE = \frac{RMSE}{[Max(y_{obs}) - Min(y_{obs})]} \quad \text{Optimal value} = 0 \tag{7}$$

$$SI = \frac{RMSE}{\overline{y_{obs}}} \quad \text{Optimal value} = 0 \tag{8}$$

$$MAPE = \frac{1}{n} \sum \left| \frac{y_{obs} - y_{calc}}{y_{obs}} \right| \quad \text{Optimal value} = 0 \tag{9}$$

$$Bias = \frac{\sum (y_{calc} - y_{obs})}{n} \quad \text{Optimal value} = 0 \tag{10}$$

$$R^2 = \left[ \frac{\sum (y_{obs} - \overline{y_{obs}})(y_{calc} - \overline{y_{calc}})}{\sqrt{\sum (y_{obs} - \overline{y_{obs}})^2 \sum (y_{calc} - \overline{y_{calc}})^2}} \right]^2 \quad \text{Optimal value} = 1 \tag{11}$$

$$NSE = 1 - \frac{\sum (y_{obs} - y_{calc})^2}{\sum (y_{obs} - \overline{y_{obs}})^2} \quad \text{Optimal value} = 1 \tag{12}$$

$$VAF = 1 - \frac{var(y_{obs} - y_{calc})}{var(y_{obs})} \quad \text{Optimal value} = 1 \tag{13}$$

$$AIC = n \times \ln(RSS) + 2k \quad \text{Optimal value} =$$
$$RSS = \sum (y_{calc} - (ay_{obs} + b))^2 \text{ the smaller the better} \tag{14}$$

where $y_{obs}$ is the observed value, $y_{calc}$ is the interpolated value by the model, $\overline{y_{obs}}$ is the mean of observed values, $\overline{y_{calc}}$ is the mean of calculated values, $n$ is the amount of data, $k$ is the number of model parameters, and $a, b$ are fit line coefficients.

The overall flow of the interpolation process is illustrated in Figure 4 for different deterministic and AI approaches.

In this study, Python version 3.8.8 was applied as the programming language. Python is an open-source and high-level language that has recently been applied widely for data analysis and machine learning. Moreover, Spyder version 5.2.2 was used as an open-source cross-platform integrated development environment for scientific programming in the Python language. The processor used was an 11th Gen Intel Core i5 @ 2.40 GHz and the installed RAM was 8.00 GB.

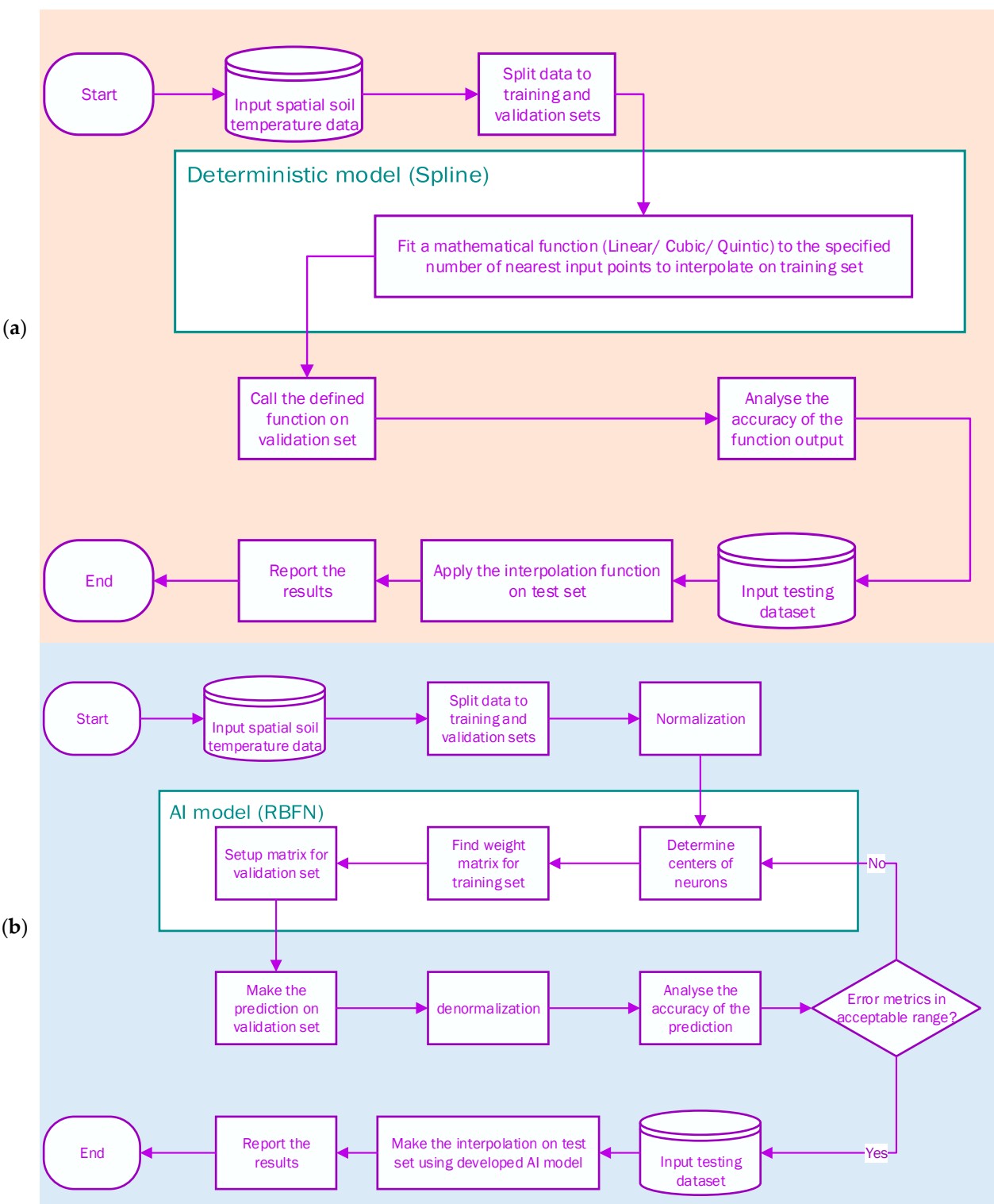

**Figure 4.** *Cont.*

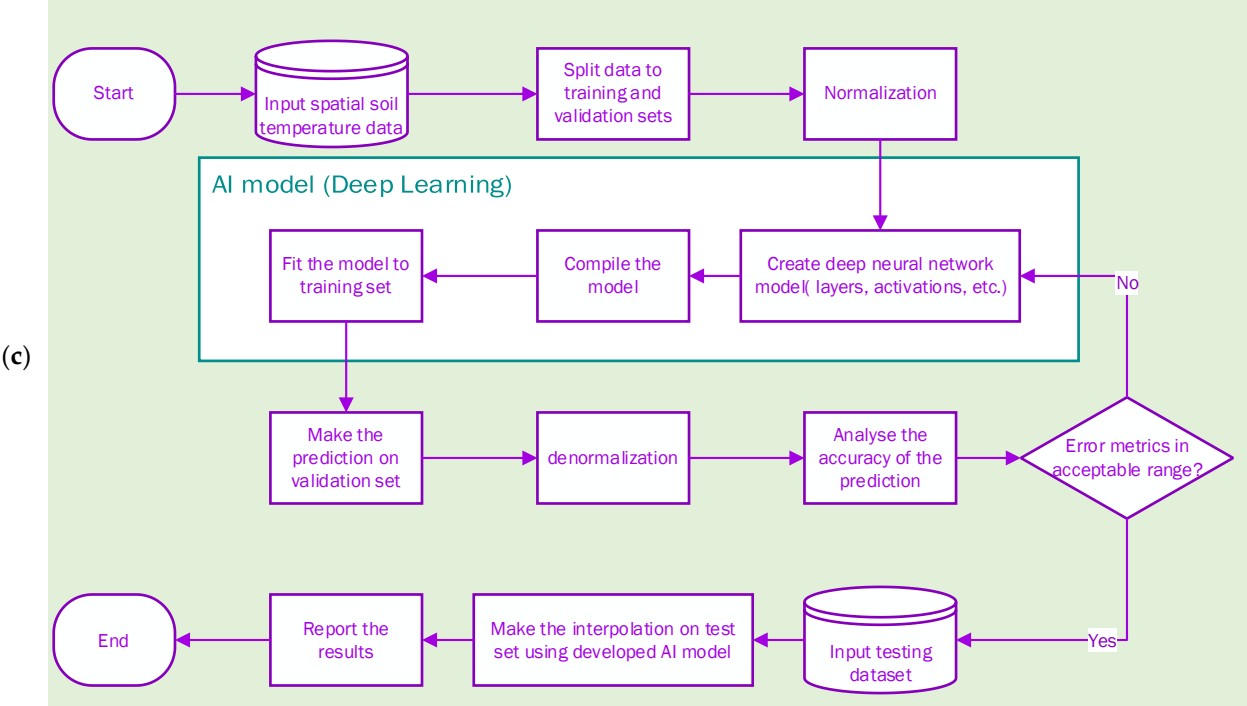

**Figure 4.** Algorithm of the interpolation models: (**a**) spline method (**b**) RBFN (**c**) deep learning.

## 3. Results

With the aim of finding the soil temperature values on the points along the railroad, each of the interpolation methods described in Section 2 was tested on the observations. The spline method was initially considered. In the first step, the function type was set to linear. The interpolated results are illustrated in Figure 5. Figure 5a shows that the linear model was not able to perform the interpolation for the whole railroad and the model outcome diverged after a while. The detailed study showed that divergence occurs from the area where soil temperature values in the reference dataset decreased sharply, and the linear interpolation model could not adapt to the data in that region. The sharp transition area has been shown with a dashed circle in Figure 5a.

In the next step, the linear interpolation was replaced by the nonlinear cubic function. The results demonstrated in Figure 5b show that the model again failed to interpolate data at the sharp changes area. Furthermore, the results of this spline model in the beginning parts of the railroad, before reaching to the sharp transition region, were very similar to linear method results, with only a 0.06% difference.

Afterward, the nonlinear spline method with a quintic function was set as the interpolation scheme. However, this more complicated spline method could not improve the performance of the interpolation model in the region of the sudden change, which is depicted in Figure 5c. It has been illustrated in the quintic 3D plot in Figure 5d that the results of interpolation diverged near the aforementioned zone. This model showed a 1.62% difference in interpolation results with linear method results for the points on the railroad before the sharp transition area.

A closer view reveals that in the area of sharp temperature transition, the railroad is passing through the water-land interface, which is also the line between high and low soil temperature. This area includes the segment of the railroad from Kingston (44.3° N, 76.5° W) to Hamilton (43.3° N, 79.9° W) that has been shown with a dashed circle in Figure 2.

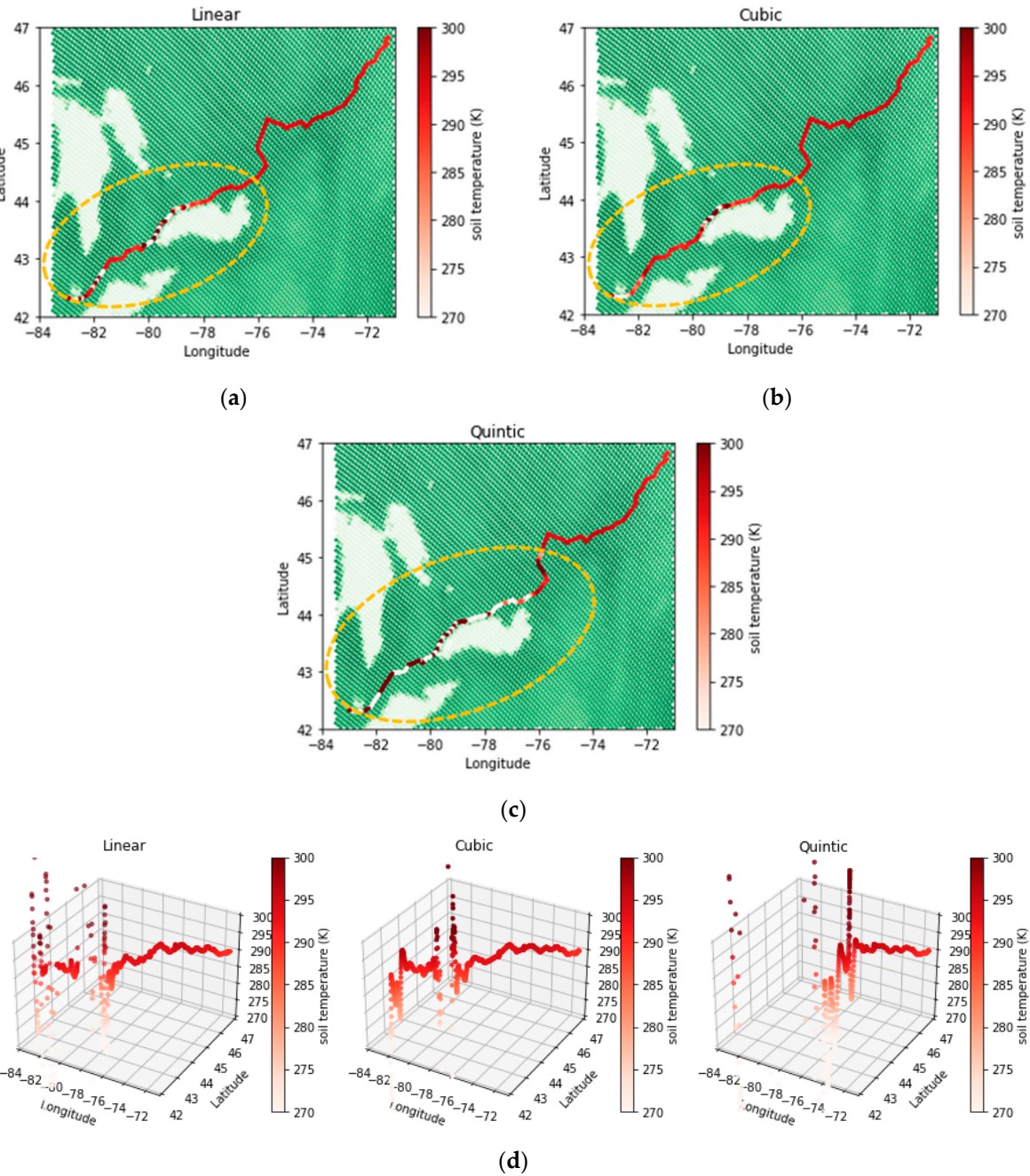

**Figure 5.** Interpolated soil temperature calculated by (**a**) linear method (**b**) cubic spline method (**c**) quintic spline method (**d**) 3D graphs for spline results.

Knowing the basic assumption of the spline interpolation method reveals that it fits a minimum curvature surface through the known interpolating points [7]. Hence, it may not be the best method of interpolation when there are large variations over relatively short horizontal distances [6]. This fact can explain the failure of the spline method in soil temperature interpolation on the coastline railroad.

The poor performance of the used methods, especially in sharp transitions, motivated the effort to try ML techniques to interpolate the gridded data to the desired locations. One of the strengths of ML is its flexibility and its not being restricted to linear relations, as in regression kriging and kriging with external drift [12]. Therefore, geostatistical approaches were discarded in this study.

An RBF neural network, with the details presented in Section 2.2.2, was developed to interpolate the soil terameter values along the railroad. The model was applied to the training set (65% of the reference dataset) to train the RBF neural network. Next, the trained model was applied to the validation set, which is the remaining part of the reference dataset. The error analysis results presented in Table 2 show that this neural network approach can interpolate data on the desired points.

**Table 2.** Error analysis of interpolated soil temperature values using different AI models.

| Error index | MaxE | MAE | MSE | RMSE | NRMSE | RRMSE |
|---|---|---|---|---|---|---|
| **Method** | (K) | (K) | ($K^2$) | (K) | (-) | (-) |
| **RBFN** | 14.89 | 2.58 | 16.50 | 4.06 | 16.25% | 1.41% |
| **Deep Learning** | 8.13 | 1.63 | 5.12 | 2.26 | 9.05% | 0.78% |
| **Error index** | MAPE | Bias | $R^2$ | NSE | VAF | AIC |
| **Method** | (-) | (K) | (-) | (-) | (-) | |
| **RBFN** | 0.90% | 0.08 | 53.81% | 53.78% | 53.80% | 23100 |
| **Deep Learning** | 0.57% | 1.13 | 89.24% | 85.65% | 89.22% | 20800 |

In addition, a deep neural network was used to develop the interpolation model, with the details presented in Section 2.2.3. The same procedure of training and validation was carried out for the deep learning model. The results of the error analysis are presented in Table 2, which shows better performance in comparison with the RBFN model in almost all error indicators.

The MAE delivers the average absolute magnitude of the errors without considering their direction. Table 2 shows that the RBFN model has higher MAE and therefore less accuracy in interpolation than deep learning. MSE is another way of delivering an average magnitude of errors. In MAE all of the individual differences are weighted equally in the average, while MSE gives a relatively high weight to large errors. Higher MSE for the RBFN model shows that this model had more large errors than the deep learning model.

Since the MSE has a square of the original data's unit, RMSE with the same unit is a more suitable metric to compare the results. The MAE and the RMSE can be used to diagnose the errors' variation. The greater difference between them, the greater the variance in the individual errors. Given that the difference between RMSE and MAE is 1.48 and 0.63 Kelvin for the RBFN and deep learning methods, respectively, the RBFN model interpolated with the larger error variance.

NRMSE and RRMSE are indices that are derived from RMSE but are dimensionless, which gives a better understanding for comparison. Table 2 shows that deep learning has a better situation from this point of view.

MAPE is a common dimensionless metric that is not sensitive to outliers and gives a general indication of model performance. It can be seen in Table 2 that deep learning performed better with lower MAPE.

In contrast with the above-mentioned indices, Bias does not determine model precision, rather it indicates the overall direction of the errors. When the Bias is a positive number, this means that the estimation was over-forecasting, while a negative number suggests under-forecasting. If the result is zero, then no bias is present. Table 2 shows that deep learning has an upward bias, while the RBFN is unbiased.

NSE is a normalized statistic that determines the relative magnitude of the residual variance compared to the actual data variance. Table 2 shows that the NSE of deep learning is closer to one than the NSE of RBFN, therefore deep learning could estimate with less error variance. VAF is another index that indicates the relative variance of model errors. In the situation of a perfect model with an estimation error variance equal to zero, the resulting VAF equals 1. Therefore, deep learning outperformed RBFN according to Table 2.

AIC is an estimator of prediction error that estimates the quality of each model, relative to each of the other models. Thus, AIC provides a means for model selection. Table 2 shows a lower value for deep learning that confirms its superiority against the RBFN model.

Scatter plots of the interpolated soil temperatures on the validation set computed by the two AI methods are illustrated in Figure 6. In this figure, the identity line is illustrated in solid orange line, error lines (here, 5% error lines) are presented in dotted orange lines, and the regression line is depicted in a dashed navy line. The identity line or line of equality is a line with a slope of 1 and is normally used in comparing two sets of data expected to be identical under ideal conditions. In this case, closer points to the identity line mean the interpolated values computed by each model are closer to the actual data. The regression line or trend line is a statistical tool that depicts the correlation between two sets of data. In the present case, the presented correlation equation shows how interpolated values and actual data correspond to each other. In general, the more the regression line matches the identity line, the closer the calculated values are to the actual data.

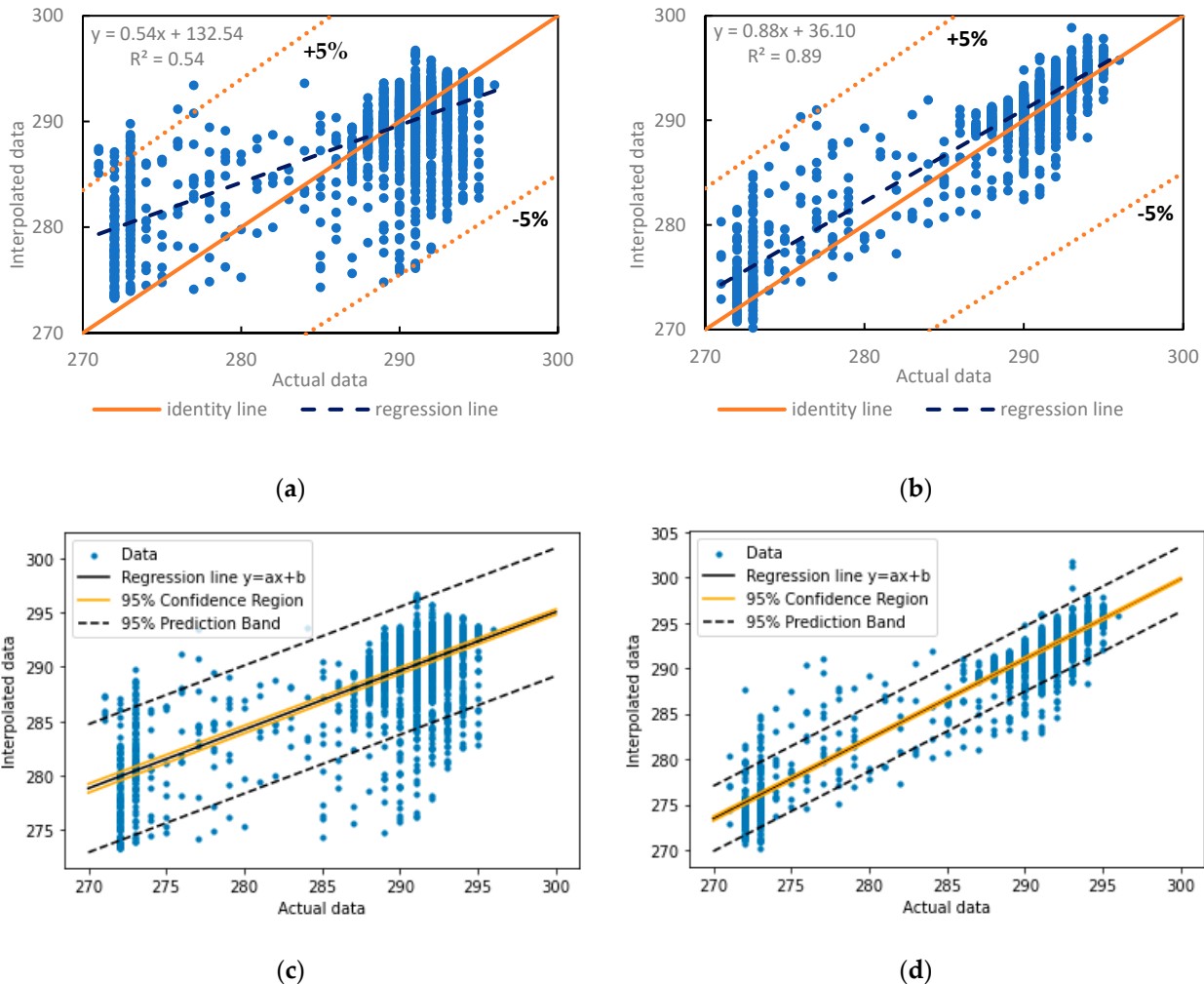

**Figure 6.** Scatter plots of interpolated and actual soil temperature, identity, and regression lines of (**a**) RBFN (**b**) Deep learning; confidence and prediction bands of (**c**) RBFN (**d**) Deep learning.

It can be seen that although most of the points are near the identity line in both Figure 6a,b, Figure 6a has more scattered points. Moreover, some points in Figure 6a are out of the 5% error lines, while all the points in Figure 6b are within the 5% error lines. Furthermore, the points of Figure 6a follow a trend which is different from the identity line (slope of 0.54 against 1), while the points of Figure 6b have a trend line quite like the identity line (slope of 0.88 against 1).

It has been shown that the identity line and regression line are relatively closer in Figure 6b compared to Figure 6a. Therefore, Figure 6b demonstrates a better match between the actual values and the models' interpolation. It was determined that the deep learning model was able to provide more reliable soil temperature interpolation than the RBFN model.

The prediction band is depicted in Figure 6c,d for deep learning and RBFN. This prediction band shows the range where one can expect the data to fall, which means it should be expected that 95% of future data points lie within the prediction bands.

A confidence band is a band around the regression line that represents the range in which the true regression line lies at 95% confidence. A narrower confidence band demonstrates a higher precision. It can be concluded from Figure 6c,d that the deep learning approach performed with more precision.

The difference between the actual and the interpolated soil temperature obtained from two developed AI models was calculated. The residuals are illustrated as box and whisker plots in Figure 7. A box-whisker plot is a standard way of demonstrating the locality and spread of residuals, displaying significant summary numbers of the minimum (lower whisker), first quartile (box lower line), median (box middle line), third quartile (box upper line), maximum (upper whisker) and mean (cross marker). The outliers are not demonstrated in this plot. The closer the residual median to zero and the smaller the residual range, the better the model performance, since this indicates that the residuals are mainly distributed around zero.

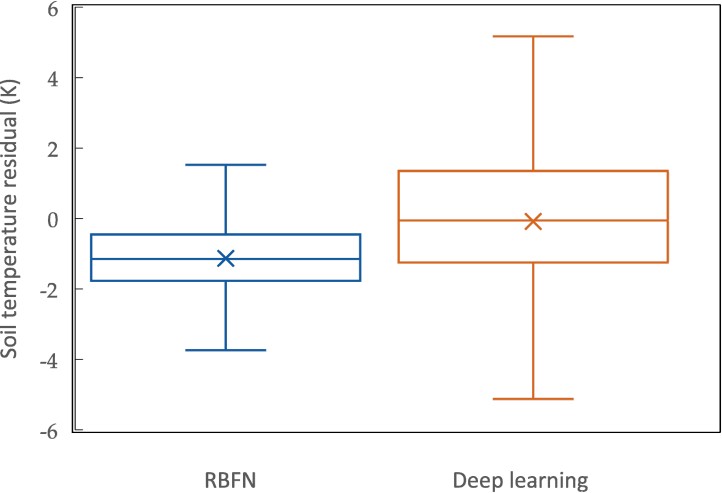

**Figure 7.** Box plot of soil temperature residuals computed by RBFN and Deep learning.

It can be seen from Figure 7 that deep learning residuals have a wider range. Also, the upper and lower whisker lengths are longer in the deep learning plot, which means that the absolute values of maximum and minimum residual calculated by this model are greater than RBFN model ones.

Moreover, the median and mean values for deep learning residuals are $-0.06$ and $-0.08$, respectively. However, the median and mean values for RBFN residuals are $-1.15$ and $-1.13$, respectively. Given that, the median of the residuals calculated by deep learning is smaller than RBFN residuals and is almost zero, which is shown in Figure 7.

It can be concluded from Figure 7 that although the distribution of residuals for the deep learning model is wider than for the RBFN model, since the mean and median of residuals computed by deep learning are less and almost zero, this model had better performance in the interpolation of the validation set.

Some more advanced models such as Convolutional Neural Networks (CNN), which consider spatial and temporal dimensions, can be investigated, and their performance can be compared with deep learning. In addition, other relevant parameters such as soil type, sub-surface type, and land characteristics should be involved in the input data, and the importance of each feature should be investigated in further studies.

## 4. Discussions

There were 6640 soil temperature pieces of data in the input database. The AI models were trained with 65% of the input database and validated with the remaining 35%. After comparing the performance of the two AI models on the validation set, the models should be applied to the testing set. The testing set in this study, as mentioned in Section 2.3, includes the locations of 1300 points along the corridor, where gridded soil temperatures need to be interpolated. Thus, the two AI models were used to interpolate soil temperature values on 1300 points with intervals of 1 km along the railroad.

The RBFN and deep learning model were applied to the testing set. The results of the soil temperature values interpolated along the railroad are presented in Figure 8.

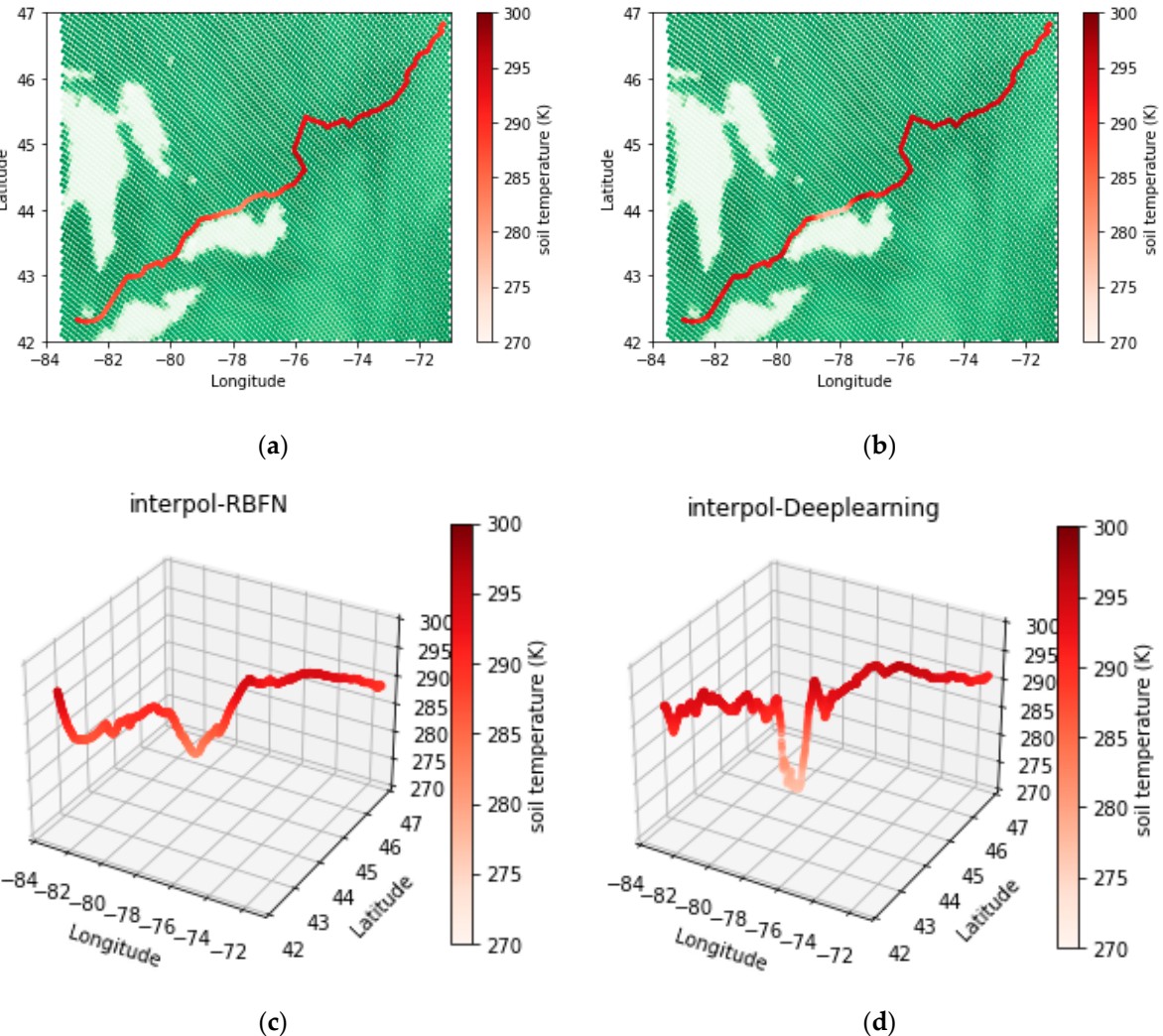

**Figure 8.** Interpolation soil temperature results along the railroad by (**a**) RBFN (**b**) Deep learning (**c**) 3D graph of RBFN (**d**) 3D graph of Deep learning.

It can be seen from Figure 8 that the developed models were able to interpolate values on the railroad even in sharp temperature transition areas. The estimated sharp transition regions can be seen in 3D graphs of the interpolated values in Figure 8c,d.

This study's objective was to obtain high-resolution soil temperatures along the railroad. Since the measured data along the railway was not available, the considered AI models were trained and validated on the ground truth gridded soil temperature data. The models were then applied to interpolate soil temperature values along the railroad. The results of the outperformed model in the validation step can be considered as more reliable interpolation results.

It was found after the validation of the models that between the considered AI techniques, deep learning offers better performance than RBFN in data interpolation, which was confirmed by Table 2 and Figures 6 and 7. Hence, considering the successful performance in interpolating the data even over sharp variations in the land-water interface area, it can be concluded that deep learning provided more reliable results for soil temperature interpolation on the corridor.

### 4.1. Interpolation of the Water Content of the Soil

With the aim of investigating the performance of two former AI models, the same study was done on the water content of the soil in the same study area. The source of the soil moisture database is detailed in Section 2.1. The RBFN and deep learning models were executed on the region depicted in Figure 3 with 6640 reference points. The models were trained with 65% of the data and validated with the remaining 35%. The details of the applied used models are presented in Sections 2.2.2 and 2.2.3, and the modeling algorithms used are explained in Figure 4.

The results of the water content error analysis are presented in Table 3. Table 3 shows that all error indices indicating precision have values closer to optimal values for the deep learning model against the RBFN model. Moreover, the Bias indicator in Table 3 shows that both models have unbiased results, which is in contrast with the soil temperature interpolation results in Table 2 that show a bias for deep learning results.

**Table 3.** Error analysis of interpolated soil water content values using different AI models.

| Error index / Method | MaxE $(kg/m^2)$ | MAE $(kg/m^2)$ | MSE $(kg^2/m^4)$ | RMSE $(kg/m^2)$ | NRMSE (-) | RRMSE (-) |
|---|---|---|---|---|---|---|
| RBFN | 0.76 | 0.10 | 0.03 | 0.17 | 17.54% | 69.55% |
| Deep Learning | 0.66 | 0.04 | 0.01 | 0.08 | 7.92% | 31.39% |

| Error index / Method | MAPE (-) | Bias $(kg/m^2)$ | $R^2$ (-) | NSE (-) | VAF (-) | AIC |
|---|---|---|---|---|---|---|
| RBFN | 48.00% | 0.00 | 56.91% | 56.88% | 56.91% | 8600 |
| Deep Learning | 20.69% | 0.01 | 91.32% | 91.21% | 91.32% | 5900 |

The water content error analysis confirmed that deep learning performs better in comparison with the RBFN model.

Afterward, the two AI models were applied to the testing set, the location of 1300 points along the corridor as described in Section 2.3. In this regard, the gridded soil water content values in the study area were interpolated along the railroad using each of the AI models. The results of the water content interpolated along the railroad are presented in Figure 9. Figure 9 shows that the RBFN and deep learning models are able to interpolate values along the railroad, even in sharp transition areas in the land-water interface region. The 3D interpolated sharp transitions can be seen in Figure 9c,d. Since the deep learning method provided more precise results in the validation step, it can be concluded that water contents interpolated along the railroad by the deep learning method are more reliable.

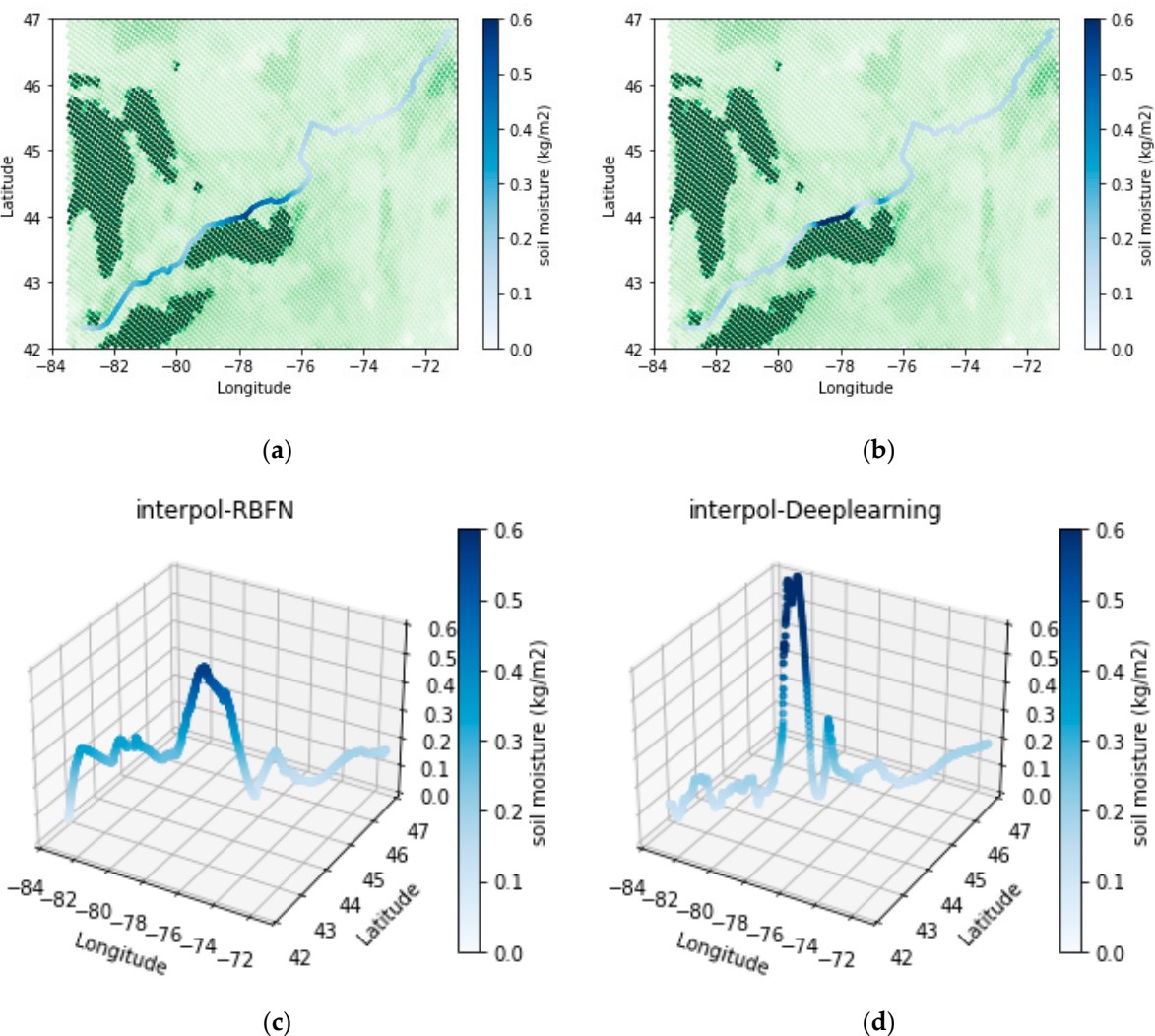

**Figure 9.** Interpolation soil water content results along the railroad by (**a**) RBFN (**b**) Deep learning (**c**) 3D graph of RBFN (**d**) 3D graph of Deep learning.

### 4.2. Evaluation of Methods' Performance along the Railroad

To assess the performance of two interpolation methods in interpolating soil temperature and water content along the railroad, the interpolated values obtained by AI models were compared with variable data at some points where information was available. The considered points are depicted in Figure 3 with red dots. The error analysis result is presented in Table 4.

**Table 4.** Error analysis of soil temperature and water content values interpolated along the railroad using different AI models.

| Variable | Soil Temperature | | Water Content | |
|---|---|---|---|---|
| **Interpolation Method** | **RBFN** | **Deep Learning** | **RBFN** | **Deep Learning** |
| RMSE | 2.26 | 1.30 | 0.09 | 0.06 |
| $R^2$ | 26% | 67% | 39% | 34% |
| Bias | −1.34 | 0.32 | 0.05 | 0.03 |

The error metrics in Table 4 show that the deep learning method interpolated soil temperature values along the railroad with smaller RMSE, higher R-squared and less absolute bias. Furthermore, the error metrics of the two AI methods in water content

interpolation were quite the same, which can be described as low RMSE, intermediate R-squared and almost unbiased.

It can thus be concluded that deep learning outperformed RBFN in soil temperature interpolation, while both deep learning and RBFN had almost good performance in water content interpolation along the railroad.

## 5. Conclusions

The need for estimating climatic parameters from globally distributed large datasets on a local site has encouraged the implementation of spatial interpolation techniques. A cost-effective model for soil temperature interpolation, which benefits from artificial intelligence techniques, is developed in the present research. Therefore, some linear and nonlinear deterministic interpolation methods in addition to the two AI methods of RBF neural networks and deep neural networks were used to generate a numerical model for soil temperature interpolation. The developed model was used to interpolate the gridded soil temperature values on the Quebec City-Windsor corridor railroad, the railway with the heaviest passenger train frequency in southeast Canada, with its 1,150 km length. The results showed that AI offers promising approaches in climate parameter interpolation, and the trained AI models demonstrated a reliable ability in soil temperature interpolation even over regions where the sharp transition in the parameters is observed, which was the main limitation faced by traditional deterministic approaches. This finding was confirmed in the same investigation on soil water content.

The key findings of this study are summarized as follows:

- The spline interpolation method, which belongs to the deterministic category, showed weaknesses in calculating interpolated values in areas with sudden variations due to its inherent property of fitting a minimum curvature surface. This limitation did not improve relatively by increasing the nonlinearity of the fitted function.
- AI methods used in this study were able to demonstrate a confident and stable performance in zones with sudden changes and can provide an alternative for deterministic interpolation methods.
- Although both RBF and deep neural networks showed successful performance in interpolating data even over sharp change areas, deep learning demonstrated overall better accuracy in validation. Therefore, interpolated temperatures estimated along the railroad, calculated with a deep neural network model, were more reliable.

**Author Contributions:** Conceptualization, A.M., H.S., J.H.C. and P.P.; methodology, H.I.; software, H.I.; validation, H.I. and A.M.; formal analysis, H.I., P.P. and A.M.; investigation, H.I.; resources, H.I. and H.S.; data curation, H.I.; writing—original draft preparation, H.I. and A.M.; writing—review and editing J.H.C., P.P., H.S. and A.M.; visualization, H.I.; supervision, A.M. and J.H.C.; project administration, A.M., J.H.C. and H.S.; funding acquisition, A.M., H.S. and J.H.C. All authors have read and agreed to the published version of the manuscript.

**Funding:** This research was funded by National Research Council Canada through the Artificial Intelligence for Logistics Supercluster Support Program, grant number AI4L-120.

**Data Availability Statement:** Parts of the data used in this manuscript are available through the corresponding author upon reasonable request.

**Conflicts of Interest:** The authors declare no conflict of interest.

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
