# Peer review of "Spatial Interpolation of Soil Temperature and Water Content in the Land-Water Interface Using Artificial Intelligence"

_water, doi:10.3390/w15030473_

Round 1

Reviewer 1 Report

This study compares the performance of conventional spational interpolation techniques with advanced AI-based techqnies for soil temprature and water content. The topic is interesting the contetn is significant. However, an English edit is required for a correct and better presentation. The follwong comments may also be considered to improve the manuscript:

- more information about the used dataset should be provided such as statistical metrics, records duration etc.

- Apllication of two AI technqiues should be justfied in a better way. As also mentioned in the manuscript there are many other techqniues such as SVR, GPR etc. than can be considered. It is also possible to add some limitations and recommendations for the study.

- Infomration about tunning the models and parameter selection can be added.

- I think Figure 3 can be removed since it is not adding considerable information to the paper.

- In Line 307, correct this sentence: "one of the interpolation methods described in Section 2 was applied on the date."

Reviewer 2 Report

See the attachment

Round 2
